# Does Bone Marrow Edema Influence the Clinical Results of Intra-Articular Platelet-Rich Plasma Injections for Knee Osteoarthritis?

**DOI:** 10.3390/jcm11154414

**Published:** 2022-07-29

**Authors:** Angelo Boffa, Alberto Poggi, Iacopo Romandini, Emanuela Asunis, Valeria Pizzuti, Alessandro Di Martino, Stefano Zaffagnini, Giuseppe Filardo

**Affiliations:** 1Clinica Ortopedica e Traumatologica 2, IRCCS Istituto Ortopedico Rizzoli, 40136 Bologna, Italy; angeloboffa@libero.it (A.B.); poggialberto8@gmail.com (A.P.); iacoporoma@gmail.com (I.R.); asunis.emanuela@gmail.com (E.A.); valeria.pizzuti@studio.unibo.it (V.P.); stefano.zaffagnini@unibo.it (S.Z.); 2Applied and Translational Research (ATR) Center, IRCCS Istituto Ortopedico Rizzoli, 40136 Bologna, Italy; g.filardo@biomec.ior.it

**Keywords:** bone marrow edema, subchondral bone, platelet-rich plasma, orthobiologics, intra-articular, injections, knee, osteoarthritis

## Abstract

Platelet-rich plasma (PRP) is increasingly used for the intra-articular treatment of knee osteoarthritis (OA). However, clinical studies on PRP injections reported controversial results. Bone marrow edema (BME) can cause symptoms by affecting the subchondral bone and it is not targeted by intra-articular treatments. The aim of this study was to investigate if the presence of BME can influence the outcome of intra-articular PRP injections in knee OA patients. A total of 201 patients were included in the study, 80 with and 121 without BME at the baseline MRI. BME area and site were evaluated, and BME was graded using the Whole-Organ Magnetic Resonance Imaging Score (WORMS). Patients were assessed with International Knee Documentation Committee (IKDC) score Knee injury and Osteoarthritis Outcome Score (KOOS) subscales, the EuroQol-Visual Analogue Scale (EQ-VAS), and the Tegner score at baseline, 2, 6, and 12 months. Overall, the presence of BME did not influence the clinical results of intra-articular PRP injections in these patients treated for knee OA. Patients with BME presented a similar failure rate and clinical improvement after PRP treatment compared to patients without BME. The area and site of BME did not affect clinical outcomes. However, patients with a higher BME grade had a higher failure rate.

## 1. Introduction

Knee osteoarthritis (OA) can be addressed by numerous conservative strategies, ranging from oral medications and crystalline glucosamine sulphate to injective therapies with corticosteroids and hyaluronic acid [1,2,3]. These options mainly provide symptom relief rather than delay the progression of cartilage degeneration, and their effectiveness is generally short-term [1].

In this scenario, orthobiologics have recently emerged as a promising option to treat knee OA with the aim to reduce symptoms, restore knee function, and possibly prevent disease progression and delay the need for metal resurfacing [1,4,5]. Numerous products developed for intra-articular treatment are currently applied in clinical practice, ranging from blood derivates to cell-based therapies [6,7,8]. Among the different orthobiologics, platelet-rich plasma (PRP) is the most investigated product, with growing literature analyzing its safety and efficacy for the intra-articular treatment of knee OA [6,9,10]. Pre-clinical studies supported a disease-modifying effect of PRP in animal models, showing a reduction in synovial inflammation and cartilage damage progression [11]. On the other hand, clinical studies on PRP injections reported controversial results.

Several studies showed satisfactory results in terms of functional improvement and reduction of pain-related symptoms, with systematic reviews and meta-analyses documenting better results for PRP when compared to saline and other injectable options, such as corticosteroids and hyaluronic acid [12,13,14,15]. Conversely, other studies reported no superiority of PRP over placebo, and current guidelines do not recommend the use of PRP in clinical practice [16,17,18,19]. These inconsistent results might be due to the availability of different PRP preparation methods, which can yield products with different compositions and characteristics and, therefore, different results, and new classifications have been recently proposed to favor a better understanding of PRP potential in the different studies [9,20]. However, besides having a common language and more standardized products, there is another aspect that may be an even more key factor leading to heterogeneous results. Bone marrow edema (BME), a common finding in joint degeneration, is currently under scrutiny [21,22]. In fact, BME can cause symptoms by affecting the subchondral bone, thus not being targeted and addressed by intra-articular treatments [23,24]. The presence of BME could therefore affect the results of PRP injections when treating OA.

The aim of this study was to investigate if the presence of BME can influence the clinical outcome of intra-articular PRP injections in patients with knee OA.

## 2. Materials and Methods

### 2.1. Study Design and Eligibility Criteria

The present study is based on the analysis of prospectively collected data from a database of patients treated with intra-articular PRP injections for knee OA between March 2009 and November 2020 (institutional ethical committee approval Prot. n. 0017366) at the IRCCS Istituto Ortopedico Rizzoli (Bologna, Italy). All patients provided written informed consent at the time of enrollment. PRP treatment was indicated for unilateral symptomatic knee OA with a history of chronic pain (at least 6 months) and/or swelling; imaging of early OA findings with cartilage degenerative signs (Kellgren–Lawrence - K-L) grade = 0, detected on magnetic resonance imaging-MRI) or OA (K–L grade 1–4); age between 18 and 80 years; no major axial deviation (varus > 5°, valgus > 5°); no focal chondral or osteochondral lesions; the absence of any concomitant knee lesion causing pain or swelling (i.e., ligamentous or meniscal injury); and the absence of hematological or cardiovascular diseases, infections, and immunosuppression. PRP procedures consisted of 1 or 3 (1-week interval) intra-articular injections of 5 mL of PRP (based on the institutional protocol available at the time of patient recruitment), which was activated with calcium gluconate, with a platelet concentration of 4 to 5 times higher than baseline whole blood values, including both PRP with and without leukocytes.

### 2.2. Patients’ Evaluation

Patients were assessed with the International Knee Documentation Committee (IKDC) and Knee injury and Osteoarthritis Outcome Score (KOOS) subscales, the EuroQol-Visual Analogue Scale (EQ-VAS), and the Tegner score at baseline and 2, 6, and 12 months after the PRP injection. Baseline variables, including age, sex, and BMI, were collected from all patients. Moreover, all participants underwent weight-bearing antero-posterior and lateral radiography to assess the baseline OA severity according to the K-L classification. Patients were included in this study based on the availability of an MRI before the injective procedure. Patients without a baseline MRI were excluded from this study. An orthopedic surgeon, blinded to the study outcome, assessed each baseline MRI for the presence or absence of BME and assigned each subject to either the BME group or the no-BME group for further evaluation. The presence of BME was confirmed on both T1 (hypo-intense signal) and T2 (hyper-intense signal) images in multiple planes. Moreover, the area size and the site (femoral condyle, tibial plateau, or patella) of the BME were determined, and the degree of BME was defined using the Whole-Organ Magnetic Resonance Imaging Score (WORMS). In detail, the BME lesions were scored from 0 to 3 based on the level of abnormality; 0 represented no signal increase, 1 represented less than or equal to 25% of the area, 2 represented 25–50% of the area, and 3 represented equal to or more than 50% [25].

Patients were considered failures if the knee needed a new injective or surgical procedure because of symptoms persisting or worsening. For these patients, the worst score value between the baseline and available follow-up evaluations was considered for the scores of the follow-ups after treatment failure.

### 2.3. Statistical Analysis

All continuous data were expressed in terms of the mean and the standard deviation of the mean, while the categorical data were expressed as frequencies and percentages. The Shapiro–Wilk test was performed to test the normality of continuous variables. The Levene test was used to assess the homoscedasticity of the data. The Repeated-Measures General Linear Model (GLM) with the Sidak test for multiple comparisons was performed to assess the differences at different follow-up times. The ANOVA test was performed to assess the between-group differences of continuous, normally distributed, and homoscedastic data, and the Mann–Whitney non-parametric test was used otherwise. The ANOVA test, followed by the post hoc Sidak test for pairwise comparisons, was performed to assess the within-group differences of continuous, normally distributed, and homoscedastic data, and the Kruskal–Wallis non-parametric test, followed by the post hoc Mann–Whitney test with Bonferroni correction for multiple comparisons, was used otherwise. The Spearman rank Correlation was used to assess correlations between numerical scores and continuous data. The Pearson Chi-square, evaluated using an exact test, was performed to investigate relationships between grouping variables. For all tests, *p* < 0.05 was considered significant. All statistical analyses were performed using SPSS v.19.0 (IBM Corp., Armonk, NY, USA).

## 3. Results

### 3.1. Patients Characteristics

The patients eligible for this study totaled 217; out of these, 201 patients were included, while 16 patients did not present one or more clinical evaluations and were considered drop-outs (7.4%). Among the included patients, there was a total of 80 patients with BME (BME group) and 121 patients without BME (no-BME group) at baseline MRI. The BME group consisted of 54 males and 26 females, with a mean age of 55.8 ± 9.9 years and a mean BMI of 27.3 ± 4.2 Kg/m^2^. The no-BME group consisted of 80 males and 41 females, with a mean age of 50.9 ± 12.1 years and a mean BMI of 25.9 ± 4.7 Kg/m^2^. Patients with BME were older (*p* = 0.003) and had a higher BMI (*p* = 0.039) compared to patients without BME. Moreover, patients with BME showed a higher severity of knee OA based on the K-L classification compared to patients without BME (*p* < 0.0005). Table 1 shows the demographic and clinical data of the included patients for each group at baseline.

### 3.2. Clinical Results

A statistically significant improvement in all clinical scores was documented from the baseline to the final follow-up for both groups (Table 2). In the BME group, the IKDC subjective score improved significantly from the baseline value of 47.7 ± 17.1 to 58.7 ± 18.9 at 2 months, 60.7 ± 20.4 at 6 months, and 63.4 ± 19.3 at 12 months (all *p* < 0.0005 vs. baseline). In the no-BME group, the IKDC subjective score improved significantly from 48.4 ± 15.8 to 57.2 ± 18.0 at 2 months (*p* < 0.0005), 60.6 ± 19.5 at 6 months (*p* < 0.0005), and 62.7 ± 20.2 at 12 months (all *p* < 0.0005 vs. baseline) (Figure 1). Similarly, the KOOS subscales and the EQ-VAS score improved from baseline to all follow-ups in both groups, as reported in detail in Table 2. The Tegner score showed a significant improvement from pre-treatment to the final follow-up, without regaining the same pre-symptoms level for both groups.

The comparative analysis between the BME group and the no-BME group did not show any significant differences in terms of absolute values and clinical improvement in all scores at all follow-ups (Table 2). Similarly, no statistically significant difference was found between the BME group and the no-BME group in terms of failures: 7 patients (8.8%) in the BME group and 12 patients (9.9%) in the no-BME group required new injective or surgical treatment during the study period.

### 3.3. BME Sub-Analysis

The BME was located at the level of the tibia in 38 patients, the femur in 16 patients, and the patella in 5 patients, while 21 patients had simultaneous lesions of the tibia and femur. The BME location did not influence the clinical outcome after PRP injections. The size of the area of the BME lesions did not affect the clinical outcomes. Similarly, the grade of BME lesions according to the WORMS (12 patients had grade 3, 24 had grade 2, and 44 had grade 1) did not influence the clinical scores after PRP treatment, although patients with grade 3 BME had a higher failure rate (25.0%) vs. patients with grade 2 (8.3%, *p* = 0.173) and patients with grade 1 (4.5%, *p* = 0.028).

## 4. Discussion

The main finding of this study is that the presence of BME at the baseline MRI did not influence the clinical results of intra-articular PRP injections in these patients treated for knee OA. Patients with BME presented a similar failure rate and clinical improvement after PRP treatment compared to patients without BME. The area and site of BME did not affect clinical outcomes. However, patients with a higher BME grade had a higher failure rate.

Intra-articular PRP injections aim to positively modulate the joint environment, thanks to the anabolic and anti-inflammatory properties of PRP due to growth factors and cytokines released by platelets [26,27]. However, the intra-articular delivery of PRP cannot address structures beyond the articular surface, such as subchondral bone, which is often involved in knee OA disease [21,22,28]. This could explain the limited and often unsatisfactory clinical results obtained by intra-articular knee injections, with their temporary effect and the inability to arrest the underlying OA process [10,16,29]. Based on this rationale, the idea to directly target the subchondral bone with orthobiologics is gaining increasing interest in clinical practice as a minimally invasive procedure to better address knee OA [30].

Subchondral bone injections with orthobiologics demonstrated promising results both in terms of clinical and imaging findings [31,32,33,34]. Nevertheless, evidence on subchondral injections is still limited and characterized by only a few high-level studies. Among these, a recent single-blinded RCT conducted by Barman et al. [35] investigated the combined use of intra-articular and subchondral PRP injections for the treatment of 50 patients with knee OA. Although the combined approach provided a greater reduction in VAS pain at 6 months compared to the intra-articular injection alone, subchondral PRP injections did not show any clinical benefit over intra-articular PRP injections alone in the other scores at the short-term follow-up, questioning the real usefulness of the subchondral injections. On the other hand, other authors reported positive findings by targeting the subchondral bone with PRP. Sanchez et al. evaluated subchondral PRP injections combined with intra-articular PRP injections for the treatment of knee OA, reporting the safety and effectiveness of this procedure, with a relatively low rate of conversion to joint replacement [36,37]. More recently, the same authors documented a superior clinical outcome at 6 and 12 months for the combined subchondral and intra-articular PRP injections when compared to intra-articular injections alone in 60 patients with severe knee OA, supporting the benefit of also directly targeting the subchondral bone area [31].

Regardless of the controversial efficacy demonstrated by different studies on treatment approaches aimed at directly injecting the subchondral bone, the role of subchondral bone in pain genesis in patients with knee OA has been confirmed by several authors. Creamer et al. [38] injected intra-articular anesthetic in OA knees and found that only 6 of 10 persons with painful OA had pain relief, suggesting that the structures responsible for pain are not always in contact with the intra-articular environment. Knee pain in OA patients can also originate from extra-articular structures, one of the most likely being the subchondral bone. In these patients, BME lesions are often considered to be responsible for knee pain. Felson et al. [23] were the pioneers of the concept of pain being associated with subchondral bone lesions. In an observational study on 401 knee OA patients, these authors demonstrated that the presence of bone marrow lesions is the strongest predictor of pain. Their study also documented that bone marrow lesions were more frequent in patients with higher K-L grades, which is in line with the results of the current study, where patients with BME showed a significantly higher severity of knee OA compared to patients without BME. Felson et al. also demonstrated, in a longitudinal study on 330 knee OA patients, that the development of knee pain is associated with an increase in bone marrow MRI lesions, while the decreasing size of bone marrow lesions is associated with the resolution of knee pain [24].

It is paramount to consider that BME is not a constant finding, as also demonstrated by Kornaat et al. [39]. in a total of 182 patients with knee OA, the authors found that BME fluctuated in most knees over a 2-year period, indicating that BME is part of a dynamic process in OA. These fluctuations may reflect the pain trajectories in knee OA, where patients experienced pain worsening or reduction over time [40]. Kornaat et al. [39] also reported that 10% of BME lesions disappeared completely over time. These fluctuations complicate the assessment of a clinical correlation between symptoms and the presence of BME. This important aspect might also partially explain the clinical improvement found in patients with BME treated with intra-articular PRP injections in the current study. In fact, the reduction in the average symptoms could be related to patients with a BME reduction over time, but unfortunately, the trajectory of the BME lesions was not evaluated with post-treatment MRI in this study.

The clinical improvement found in patients with BME after intra-articular PRP treatment could also be related to a possible effect at the subchondral bone level of the intra-articular delivery of PRP [11]. In fact, some preclinical studies suggested that intra-articular PRP can also provide disease-modifying effects in the subchondral bone, with a lower number of cyst formations and less severe edematous changes compared to OA controls [41,42]. Moreover, one clinical study reported a significant reduction in bone marrow lesions at 3 and 6 months after six intra-articular PRP injections [43]. However, this study was a case series of only 50 knees and lacked a placebo control, and other clinical evidence suggesting that intra-articular injections of orthobiologics could favor a direct improvement of the subchondral bone is limited. Therefore, future high-level studies should evaluate the possible benefits of intra-articular PRP injections on the subchondral bone in the clinical setting, as well as the possible influence of different types of BME lesions. Some BME lesions may be transitory, but larger lesions have been correlated with a worse outcome and a higher risk of arthroplasty [44]. This could also explain the results of the current study. Most BME lesions were of low grade, and no overall BME influence was found on clinical outcomes. However, patients with a higher BME grade also presented a higher failure rate. Thus, it is possible that different results could be found when investigating PRP treatment for more severe OA patients.

This study presents some limitations that should be considered in the interpretation of the results. The PRPs used were heterogeneous and only had partial characterization of the injected products. Future studies should better characterize the PRP used to better understand if this could influence the observed results, as also recently underlined by a consensus of experts suggesting an in-depth coding system for PRP studies [9]. The absence of MRI evaluations performed over time did not allow us to assess the evolution of BME. Therefore, the correlations of possible BME fluctuations or resolution with the clinical outcome were not investigated. There is an age difference between the BME and no-BME groups, which could influence the results of this study. Interestingly, despite a significantly higher age in patients with BME, comparable overall clinical results with respect to the no-BME group were observed. Age can have various effects on the final results. Patients with a too-advanced age do not respond well to these orthobiologic treatments, as previously reported [45]. On the other hand, too-young patients also seem to only gain a partial benefit from these injections, likely due to their higher activity level and therefore higher expectations [10]. In this light, the results of PRP injections seem to be influenced by age in a complex and nonlinear way, thus warranting caution when comparing non-homogenous groups of patients. The relatively small sample size of patients with BME might have hindered the detection of significant factors that could influence the results based on BME characteristics, in particular considering the low representation of more severe BME grades. This is also true for subgroup analyses of other factors, such as the number of injections received or other patient/treatment-related characteristics. It is also possible that, since BME may be associated with different grades of chondromalacia, this could influence the overall results. Finally, a larger series of patients would have allowed us to perform a match-paired analysis to better understand the role of high BME levels among more homogeneous patient cohorts. Therefore, future studies with a large number of patients with BME should analyze the response to intra-articular injections and the possible correlation of several variables with the clinical outcomes. The identification of specific BME lesion types and grades influencing the clinical outcome could optimize the indication for PRP intra-articular injections in the treatment of knee OA.

## 5. Conclusions

This study demonstrated that the presence of baseline BME at MRI did not affect the clinical outcome of intra-articular PRP injections in OA patients. A comparable failure rate and clinical improvement were found between patients with and without BME. However, patients with a higher BME grade showed a higher failure rate of PRP treatment for knee OA.

## Figures and Tables

**Figure 1 jcm-11-04414-f001:**
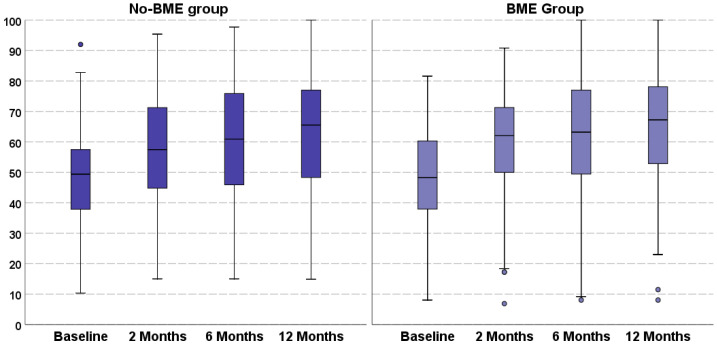
The International Knee Documentation Committee subjective score trend in both no-BME and BME groups. The box-and-whisker plots show median, quartile, and 95% confidence interval.

**Table 1 jcm-11-04414-t001:** Baseline demographic characteristics and clinical scores of patients of both groups.

	BME Group (n = 80)	No-BME Group (n = 121)	*p* Value
**Sex (Male/Female)**	54 / 26	80 / 41	n.s.
**Age, y (mean ± SD)**	55.8 ± 9.9	50.9 ± 12.1	*p* = 0.003
**BMI, kg/m^2^**	27.3 ± 4.2	25.9 ± 4.7	*p* = 0.039
**Kellgren-Lawrence Grade**	Grade 1: 0	Grade 1: 24	*p* < 0.0005
	Grade 2: 39	Grade 2: 67	
	Grade 3: 30	Grade 3: 27	
	Grade 4: 11	Grade 4: 3	
**IKDC Subjective**	47.7 ± 17.1	48.4 ± 15.8	n.s.
**Tegner Pre-Symptoms**	4.3 ± 2.0	4.8 ± 1.8	n.s.
**Tegner Pre-Treatment**	2.7 ± 1.5	2.9 ± 1.3	n.s.
**KOOS Pain**	63.7 ± 17.9	66.4 ± 16.6	n.s.
**KOOS Symptom**	63.0 ± 18.5	65.3 ± 16.8	n.s.
**KOOS ADL**	70.6 ± 20.9	73.3 ± 17.6	n.s.
**KOOS Sport/Rec**	40.2 ± 24.2	44.5 ± 23.1	n.s.
**KOOS QOL**	37.1 ± 17.4	35.4 ± 18.3	n.s.
**EQ-VAS**	68.7 ± 16.5	71.5 ± 15.0	n.s.

Clinical data are expressed as mean and standard deviation (SD). BMI, body mass index; n.s., not significant; y, years.

**Table 2 jcm-11-04414-t002:** Clinical scores during follow-up in the BME and no-BME groups.

Outcome	Group	Baseline	2 Months	6 Months	12 Months
**IKDC Subjective Score**	*BME group* *No-BME group*	47.7 ± 17.1 48.4 ± 15.8	58.7 ± 18.9 * 57.2 ± 18.0 *	60.7 ± 20.4 * 60.6 ± 19.5 *	63.4 ± 19.3 * 62.7 ± 20.2 *
**KOOS Pain**	*BME group* *No-BME group*	63.7 ± 17.9 66.4 ± 16.6	73.0 ± 18.9 * 74.4 ± 16.3 *	75.9 ± 18.6 * 74.6 ± 18.2 *	77.9 ± 19.0 * 77.6 ± 18.0 *
**KOOS Symptoms**	*BME group* *No-BME group*	63.0 ± 18.5 65.3 ± 16.8	72.8 ± 20.0 * 72.1 ± 15.1 *	75.6 ± 19.8 * 73.7 ± 16.5 *	76.6 ± 19.3 * 74.4 ± 17.3 *
**KOOS ADL**	*BME group* *No-BME group*	70.6 ± 20.9 73.3 ± 17.6	80.8 ± 18.0 * 80.8 ± 16.8 *	82.1 ± 18.0 * 80.7 ± 17.8 *	83.6 ± 18.9 * 82.6 ± 17.7 *
**KOOS Sport/Rec**	*BME group* *No-BME group*	40.2 ± 24.2 44.5 ± 23.1	49.6 ± 26.4 * 55.3 ± 24.1 *	54.1 ± 26.5 * 58.0 ± 24.1 *	57.3 ± 27.2 * 58.0 ± 25.9 *
**KOOS QoL**	*BME group* *No-BME group*	37.1 ± 17.4 35.4 ± 18.3	49.4 ± 21.1 * 47.0 ± 21.7 *	53.3 ± 24.3 * 50.7 ± 23.9 *	57.5 ± 24.5 * 55.5 ± 24.1 *
**EQ-VAS Score**	*BME group* *No-BME group*	68.7 ± 16.5 71.5 ± 15.0	75.7 ± 13.9 * 74.6 ± 13.6	75.9 ± 16.1 * 77.2 ± 12.1 *	76.5 ± 16.3 * 77.6 ± 13.5 *
**Tegner Score**	*BME group* *No-BME group*	2.7 ± 1.5 2.9 ± 1.3	3.3 ± 1.6 * 3.6 ± 1.5 *	3.5 ± 1.8 * 3.7 ± 1.6 *	3.5 ± 1.6 * 3.8 ± 1.5 *

* Statistically significant improvement (*p* < 0.05) from baseline to the evaluated follow-up. No intergroup significant differences were observed in all scores at all follow-ups. ADL, Activities in daily living; BME, Bone Marrow Edema; EQ-VAS, EuroQol-visual analogue scales; KOOS, Knee injury and Osteoarthritis Outcome Score; IKDC, International Knee Documentation Committee Subjective score.

## Data Availability

Not applicable.

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
