# Peer review of "Does Bone Marrow Edema Influence the Clinical Results of Intra-Articular Platelet-Rich Plasma Injections for Knee Osteoarthritis?"

_jcm, 2022, doi:10.3390/jcm11154414_

Round 1

Reviewer 1 Report

Dear Authors, 

well-done study, I have some points, which should be cleared. 

First of all - the study design. You collected data since 2009 - so how many patients to you include - how many were excluded. 
You write that the study is prospective - so why are the groups are inhomogeus. 

For answering your scientif question you need to preform a matched-group analyses. 

Please clarify this - 

In your conlusion you sum up, that highe BME grade had a higher failure rate - but otherwise this not statistically - so please overwork you conclusion

Author Response

Response to Reviewer 1 Comments

Dear Authors,

well-done study, I have some points, which should be cleared.

Dear reviewer,

thank you for your positive comments and for your revision. We modified our manuscript following your suggestions and advice. Please see the following point by point answers to your comments and the related changes.

Point 1: First of all - the study design. You collected data since 2009 - so how many patients to you include - how many were excluded.

Response 1: This study is based on the analysis of prospectively collected data from a database of patients treated with intra-articular PRP injections at our Institute. The included patients do not represent the entire prospective cohort of our database, since we included only patients with a baseline MRI, which was the focus of the study: investigate if the presence of bone marrow edema can influence the clinical outcome of intra-articular PRP injections in patients with knee OA. Patients without a baseline MRI were not considered eligible for this study. We now better specified this aspect in the method section (lines 92-93). So, with regard to the study focus, among patients considered eligible for this study, we had a database of 217 patients: out of these, 201 presented all clinical evaluations at all follow-ups and were included in the analysis, while 16 patients did not present one or more clinical evaluation and for this reason were considered as drop-out (rate 7.4%). We now clarified this aspect in the result section (lines 127-130).

Point 2: You write that the study is prospective - so why are the groups are inhomogeus. For answering your scientif question you need to preform a matched-group analyses. Please clarify this –

 Respone 2: Thank you for your insight, it is an interesting yet challenging aspect within such exploratory analysis. This study is based on the analysis of patients treated with PRP injections at our Institute and prospectively evaluated. We decided to include all patients with a baseline MRI and compare all patients with BME with patients without BME. We considered the obtained inhomogeneity between the two groups an important result of our study. In fact, it is important to underline that patients with BME had older age, higher BMI, and higher OA degree than patients without BME. This has been reported also in previous studies, with BME being more frequent in older patients and in patients with a more severe OA. This consistency with the literature confirms the representativeness of our cohort and the generalizability of the study findings. Finally, we agree it would be interesting to have a match-paired analysis, but this would require a higher number of patients, which were not enough in this cohort (only 41 more patients in the non-BME group to pick from to match the BME group of 80 patients). Anyway, since this is an interesting aspect and could be useful to encourage future studies in this direction, we discussed these aspect in the discussion section (lines 236-239 and lines 276-293).

Point 3: In your conlusion you sum up, that highe BME grade had a higher failure rate - but otherwise this not statistically - so please overwork you conclusion

Response 3: Patients with higher BME (grade 3) showed a higher failure rate (25%) compared to other groups, with a statistically significant difference compared to patients with lower BME (grade 1, 4.5% – p = 0.028). We underlined this aspect in the results section and the conclusions have been written accordingly (lines 188-189).

Reviewer 2 Report

Interesting article on a little-known topic. I have a few comments: in the introduction and discussion, please discuss the use of hyaluronic acid injection and the oral use of crystalline glucosamine sulphate in gonarthrosis. in the discussion, please discuss the possible influence of the age difference between the groups on the results.

Author Response

Response to Reviewer 2 Comments

English language and style are fine/minor spell check required.

Interesting article on a little-known topic.

Dear reviewer,

Thank you for recognizing the relevance of this study.

We sent the manuscript for the correction of the English language to a professional English translator for the editing service.

I have a few comments:

Point 1: in the introduction and discussion, please discuss the use of hyaluronic acid injection and the oral use of crystalline glucosamine sulphate in gonarthrosis.

Response 1: We discussed the use of hyaluronic acid injection and the oral use of crystalline glucosamine sulphate as conservative treatment for knee OA in the introduction section (lines 31-35). We also added a pertinent reference. We focused this part in the introduction section rather that in the discussion section, where we focused more on the results of our study on PRP and bone marrow edema.

Point 2: in the discussion, please discuss the possible influence of the age difference between the groups on the results.

Response 2: We discussed the possible influence of the age difference between the groups on the results in the discussion section, with some related pertinent references, as requested by the reviewer (lines 276-285).

Reviewer 3 Report

Dear authors,

I find the research article here presented to be well-written and scientifically sound. I found no need for English language editing.

The one problem I have is with the results section. Even though all of the aspects of the conclusions are presented, the materials and methods section state that the patients who were enrolled in the study were derived from three groups (lines 65-72). This is not further analyzed in the results or discussed in the discussion section. Also, BME is associated with a higher grade of chondromalacia observed on MR images. Therefore, I believe it would be prudent to include a subgroup analysis of the outcomes for the included patient subgroups and to include the semiquantitative score for cartilage degeneration which is also included in the WORMS method you have used. 

In this way, you could potentially come up with arguments that the presence of BME in some subgroups has a significantly higher failure rate, which would enable better clinical reasoning for PRP therapy in patients presenting with BMEs on the initial assessment and a more precise approach that our patients require.

Kind regards!

Author Response

Response to Reviewer 3 Comments

Dear authors,

I find the research article here presented to be well-written and scientifically sound. I found no need for English language editing.

Dear reviewer,

Thank you for recognizing the relevance of this study.

Please find below our answers and the reference to the changes we made in text.

Point 1: The one problem I have is with the results section. Even though all of the aspects of the conclusions are presented, the materials and methods section state that the patients who were enrolled in the study were derived from three groups (lines 65-72). This is not further analyzed in the results or discussed in the discussion section.

Response 1: Thank you for this consideration, which allowed us to better check if we addressed this aspect correctly. In the database used for this study we included patients previously treated with a single or three injections of PRP at our Institute, based on the different protocols available over time (not based on patients, there was no different indication for patient and treatment) and then prospectively followed and evaluated with clinical evaluations. As requested by the reviewer, we analyzed patients based on the number of injections received. The groups do not differ in terms of edema presence at baseline MRI. Thus, we can confirm this aspect does not seem to influence the results which are the focus of this study. On the other hand, it is also possible that we would need more patients to underline more differences among subgroups, thus we added this aspect at the end in the limitation section, to raise awareness on this aspect as correctly suggested by the reviewer.

Point 2: Also, BME is associated with a higher grade of chondromalacia observed on MR images. Therefore, I believe it would be prudent to include a subgroup analysis of the outcomes for the included patient subgroups and to include the semiquantitative score for cartilage degeneration which is also included in the WORMS method you have used. In this way, you could potentially come up with arguments that the presence of BME in some subgroups has a significantly higher failure rate, which would enable better clinical reasoning for PRP therapy in patients presenting with BMEs on the initial assessment and a more precise approach that our patients require.

Response 2: This is an important expert-based insight, we agree, and we thank you for that. We tried to address it by asking to a professional independent statistician. This analysis presents a similar problem as the previous attempt to perform further sub-investigations. Sub-investigations per se imply the focus on smaller subgroups than what identified as primary outcome, which implies the risk of being underpowered. This is particularly true for this aspect. WORMS present an evaluation of 9 scales on 14 anatomical sublocations. Even considering only the scale on the parameters focused on cartilage, and even focusing only on the area where BME has been identified, there are many subgroups, namely 8 subgroups. Dividing the subgroup of patients with BME for these 8 subgroups would create really small subgroups. Our statistician advised against such analysis, since when the subgroups are too small it is easy to have misleading results and conclusions. However, since we find this aspect to be interesting and worthy of attention, we added it in the discussion and underline the need for further studies on larger populations to investigate this possible relationship (lines 288-293).

Round 2

Reviewer 1 Report

Dear Authors, 

good overwork. 

Author Response

Dear reviewer,

Thank you for appreciating the efforts made to improve our manuscript.

We think the new manuscript is now more interesting for the readers and suitable for publication in JCM.